# Comparative Analyses of Plastomes of Four *Anubias* (Araceae) Taxa, Tropical Aquatic Plants Endemic to Africa

**DOI:** 10.3390/genes13112043

**Published:** 2022-11-05

**Authors:** Li Li, Changkun Liu, Kunpeng Hou, Wenzhe Liu

**Affiliations:** 1School of Life Science, Northwest University, Xi’an 710069, China; 2CAS Key Laboratory for Plant Diversity and Biogeography of East Asia, Kunming Institute of Botany, Chinese Academy of Sciences, Kunming 650201, China; 3Key Laboratory of Bio-Resources and Eco-Environment of Ministry of Education, College of Life Sciences, Sichuan University, Chengdu 610065, China

**Keywords:** *Anubias*, aquarium plants, plastomes, positive selection

## Abstract

*Anubias* Schott (Araceae) have high ornamental properties as aquarium plants. However, the genus has difficulties in species identification, and the mechanism of its adaptation to the aquatic environment is unknown. To better identify species and understand the evolutionary history of *Anubias*, the plastomes of *Anubias barteri* Schott, *A. barteri* var. *nana* (Engl.) Crusio, and *A. hastifolia* Engl., were sequenced. The sizes of the plastomes of *Anubias* ranged from 169,841 bp to 170,037 bp. These plastomes were composed of conserved quadripartite circular structures and comprised 112 unique genes, including 78 protein-coding genes, 30 transfer RNA genes, and 4 ribosomal RNA genes. The comparative analysis of genome structure, repeat sequences, codon usage and RNA editing sites revealed high similarities among the *Anubias* plastomes, indicating the conservation of plastomes of *Anubias*. Three spacer regions with relatively high nucleotide diversity, *trnL-CAA-ndhB*, *ycf1-ndhF*, and *rps15-ycf1*, were found within the plastomes of *Anubias*. Phylogenetic analysis, based on 75 protein-coding genes, showed that *Anubias* was sister to *Montrichardia arborescens* (L.) Schott (BS = 99). In addition, four genes (*ccsA*, *matK*, *ndhF*, and *ycf4*) that contain sites undergoing positive selection were identified within the *Anubias* plastomes. These genes may play an important role in the adaptation of *Anubias* to the aquatic environment. The present study provides a valuable resource for further studies on species identification and the evolutionary history of *Anubias*.

## 1. Introduction

The genus *Anubias* Schott (Araceae, Alismatales) consists of eight perennially herbaceous species endemic to the western and central tropical Africa [1]. *Anubias* plants tend to grow on the banks of small streams, rough rocks, or driftwoods in tropical humid forests and are sometimes completely submerged [1,2]. They can be specified as aquatic plants because, although they are not physiologically bound to water, they are able to tolerate longer periods of submergence [2,3]. They are widely cultivated as aquarium plants, owing to their aquatic life form (helophytes or rheophytes), exotic appearance and easy maintenance [2,4]. Plants of this genus have been commercialized around the world as highly demanded aquatic ornamental plants [5]. To date, numerous polymorphic cultivars have been developed, such as *A. barteri ‘marble’* and *A. barteri* var. *nana ‘petite’* [6]. Although *Anubias* species can be identified by using mostly characteristics of the inflorescence [1], the inflorescence is often not available, due to its slow growth rate [7]. Leaf blades within the genus *Anubias* are highly plastic in their morphology [1,2]. Moreover, there are still many taxonomic problems, such as incomplete species names and synonyms among the *Anubias* species, making classification inconsistent both on the markets and in aquariums [5,8]. Considering their high economic value, there is a need for further improvement of species identification within the genus.

Accurate species identification of plants is fundamental for their utility, breeding new cultivars, and implementation of conservation priorities and biosecurity monitoring. Historically, identification of the majority of plant species has been based on the analysis of morphological variation, which has resulted in low reliability of species identification due to phenotypic plasticity [9]. The analysis of DNA sequence variation can provide useful information for species identification and for phylogenetic analysis [10]. The chloroplast (cp) genomes of plants are relatively conserved in size, organization, gene content and order, giving them unique values in comparative genomics and phylogenetics, compared to the nuclear and mitochondrial genomes [11]. To date, only one cp genome (*A. heterophylla* Engl.) in the genus *Anubias* has been reported [12]. Therefore, it is necessary to acquire more cp genomes within the genus *Anubias*.

Understanding the mechanism of plant adaptations to the aquatic environment is an important topic in evolutionary biology. Aquatic plants occupy stressful water habitats characterized by low light levels, reduced carbon and oxygen availability, and mechanical damage through wave exposure [13]. To survive in the aquatic environment, aquatic plants have undergone a series of morphologic and physiologic adaptive changes. For example, the aerenchyma in roots, stems, and leaves enhances the capture and transportation of oxygen [14,15]. At the metabolic level, aquatic plants might change the level of glycolytic fluxes and ethanolic fermentation [16]. Although all of the *Anubias* species have adapted to the aquatic environment, the related mechanism has rarely been reported.

In this study, we sequenced and assembled the plastomes of *A. barteri* Schott, *A. barteri* var. *nana* (Engl.) Crusio, and *A. hastifolia* Engl; then, we compared them with the published plastome sequences of *A. heterophylla* and other genera in the family Araceae. Furthermore, the phylogenetic relationships among *Anubias* and other genera in the Arum family were reconstructed based on consensus protein-coding genes. Our main objectives were to (1) explore the size range and structure of the *Anubias* plastomes; (2) detect highly variable regions as the bases of developing molecular markers for species identification; (3) construct a phylogenetic tree for investigating the interspecific relationships within *Anubias*, as well as the relationships among *Anubias* and other genera in Araceae; (4) identify the protein-coding genes under positive selection within the four plastomes of *Anubias*.

## 2. Materials and Methods

### 2.1. Plant Material, Plastome Sequencing, Assembly and Annotation

The samples of cultivated plants, including *A. barteri* (voucher number: JYH085), *A. barteri* var. *nana* (voucher number: JYH087), and *A. hastifolia* (voucher number: JYH086), were collected by Yunheng Ji on August 27, 2018, from Kunming World Horticultural Expo Garden, Yunnan, China. The voucher specimens were deposited in the Herbarium of Kunming Institute of Botany, Chinese Academy of Sciences (KUN, Yunheng Ji, jiyh@mail.kib.ac.cn). Total genomic DNA of three *Anubias* plants was extracted from 20 mg silica gel-dried leaf tissues using the CTAB method [17]. Then, genomic DNA was fragmented into 500 bp fragments to construct a paired-end library according to the manufacturer’s protocol (Illumina, San Diego, CA, USA) and finally sequenced on the Illumina HiSeq 2000 system.

Raw reads were assembled using the software GetOrganelle v1.7.1 [18], with the default parameters, and the plastome sequence of *A. heterophylla* (GenBank accession number: MN046884) was set as a reference. The assembled genomes were annotated using the PGA (Plastid Genome Annotator) software [19] and the online software, Geseq [20]. The preliminary annotated sequences were manually corrected for start and stop codons and intron/exon boundaries in Geneious prime 2019.2.1 [21]. All the tRNA genes were further verified using the online software tRNAscan-SE, version 2.0 [22], with the default parameters. Furthermore, circular genome maps of *Anubias* were visualized using the online program OrganellarGenomeDRAW version 1.3.1 [23]. Finally, newly sequenced plastomes of *A. barteri*, *A. barteri* var. *nana*, and *A. hastifolia* were deposited into GenBank with the accession numbers OP279443, OP279444, and MW984413, respectively.

### 2.2. Comparative Plastome Analysis

To better understand the interspecific variation of the plastomes in the genus *Anubias*, one published and three newly sequenced plastomes of *Anubias* were compared. Moreover, 27 published plastomes (Appendix A) in the family Araceae were added to determine the intergeneric variation among *Anubias* and other genera. Genome rearrangements within the family Araceae, including *Anubias* and other genera, were identified using the Mauve alignment [24], after removing all the IRa regions. Interspecific variations among the complete plastomes in the genus *Anubias* were performed in the online program mVISTA [25] with the shuffle-LAGAN model, using *A. heterophylla* as a reference. The boundaries of the LSC (large single-copy), SSC (small single-copy), and IRs (inverted repeats) of the plastomes of *Anubias* and other genera were visualized with IRscope [26].

### 2.3. Repeats, Nucletide Diversity, Codon Usage and RNA Editing Sites

The REPuter program [27] was used to analyze the numbers of forward, palindromic, reverse and complement repeats of four *Anubias* plastomes with the following parameters: Hamming distance was 3, and minimal repeat size was 30 bp. The Simple Sequence Repeats (SSRs) of four *Anubias* plastomes were detected using the MIcroSAtellite identifcation tool (MISA) [28], including mononucleotide, dinucleotide, trinucleotide, tetranucleotide, pentanucleotide, and hexanucleotides, with a minimum number of 8, 4, 4, 3, 3, and 3, respectively.

Four *Anubias* plastomes were aligned using MAFFT v7.450 [29]. Nucleotide diversity analysis was performed using DnaSP v.6.12.01 software [30] with the parameters of 600 bp in window length and 200 bp in step size.

To identify the frequency of synonymous codon and codon biases within the genus *Anubias*, the relative synonymous codon usage (RSCU) [31] of the protein-coding genes was analyzed in CondoW v1.4.2 software [32], after removing the sequences less than 300 bp [33]. The RNA editing sites in the plastomes of *Anubias* were analyzed by a predictive RNA editor for plants (PREP-cp) with the default settings [34].

### 2.4. Phylogenetic Analysis

To examine the phylogenetic positions of *Anubias*, 31 aroid taxa, including 3 *Anubias* taxa, were sequenced in this study and 28 taxa in the family Araceae downloaded from the GenBank were selected in this study (Appendix A). The species *Zamioculcas zamiifolia* (Lodd.) Engl. was set as the outgroup. A maximum likelihood (ML) tree was constructed based on 75 protein-coding genes (Appendix A) shared by the 31 plastomes. Each selected protein-coding gene sequence alignment was performed with MAFFT v.7.450 and concatenated to a supermatrix. The ML tree was constructed using the RAxML-HPC2 program [35] on the XSEDE resource in the CIPRES Science Gateway [36] with the GTRGAMMA model. Bootstrap support values were obtained with 1000 bootstrap replicates.

### 2.5. Positive Selection Analysis

To identify positively selected protein-coding genes in the genus *Anubias*, each protein-coding gene sequence matrix was aligned using MAFFT v.7.450. The stop codons were again manually deleted in each aligned sequence. The phylogenetic tree for each protein-coding gene was constructed using the FastTree 2.1.11 plugin [37] of Geneious prime 2019.2.1. The ratio of nonsynonymous (dN) and synonymous substitution (dS) (ω = dN/dS) was calculated using the site-specific model (M0, M1a, M2a, M3, M7, M8, and M8a) based on likelihood ratio tests (LRTs) with statistically significant *p* values (<0.05) performed in CODEML algorithms [38] implemented in EasyCondelML v1.4 software [39].

## 3. Results

### 3.1. Plastome Features

Complete plastomes of *A. barteri*, *A. barteri* var. *nana*, and *A. hastifolia* were assembled with genome sizes of 169,910 bp, 169,929 bp, and 169,841 bp in length, respectively (Figure 1). The sizes within the plastomes of Araceae in this study ranged between 158,177 bp (*Anchomanes hookeri* Schott) and 175,906 bp (*Zantedeschia elliottiana* Engl.). Four *Anubias* plastomes were composed of a circular conserved quadripartite structure, with similar gene content and genome organization. Each sample within the *Anubias* plastomes comprised a total of 112 unique genes, including 78 protein-coding genes, 30 transfer RNA genes, and 4 ribosomal RNA genes (Table 1).

The overall GC content percentages of *A. barteri* (35.2%), *A. barteri* var. *nana* (35.2%), *A. hastifolia* (35.1%), and *A. heterophylla* (35.1%) were similar, and this number ranged between 34.7% (*Schismatoglottis calyptrata* (Roxb.) Zoll. & Moritzi) and 37.0% (*Anchomanes hookeri*) within Araceae (Appendix A). The LSC length ranged between 75,594 bp (*Anchomanes hookeri*) and 94,702 bp (*Arisaema franchetianum* Engl.), with an average length of 91,000 bp; the SSC length ranged between 8432 bp (*Zantedeschia elliottiana*) and 24,871 bp (*Pinellia peltata* C.Pei), with an average length of 20,216 bp; and the IR length ranged between 24,982 bp (*Pinellia peltata*) and 39,445 bp (*Zantedeschia elliottiana*), with an average length of 27,553 bp (Appendix A). The Mauve alignment revealed that no genome rearrangements existed within the plastomes of *Anubias,* but did exist among other 6 genera (such as *Anchomanes* Schott, *Arisarum* Mill., *Zantedeschia* Spreng.) in the family Araceae (Appendix A).

### 3.2. Contraction and Expansion of INVERTED repeats

Comprehensive comparative analysis of the junction was performed among the 31 taxa for the contraction and expansion in JLB (LSC/IRb), JSB (IRb/SSC), JSA (SSC/IRa), and JLA (IRa/LSC). Four *Anubias* plastomes shared similar junction structures. JLB junctions were located between the *rps19* gene and the *rpl2* gene in four *Anubias* plastomes and other studied Araceae plastomes, with the exception of *Zantedeschia elliottiana*, *Zomicarpella amazonica* Bogner, and *Anchomanes hookeri*, which presented the JLB within the *rpl22*, *rpl2*, *rpl23* genes, respectively. JSB and JSA junctions were relatively conserved in four *Anubias* plastomes but highly variable in other analyzed Araceae plastomes. In four *Anubias* plastomes, the JSB junctions were characterized by the presence of a truncated copy of the *ycf1* gene. This phenomenon was also found in seven other plastomes (*Philodendron lanceolatum* Schott, *Homalomana occulta*, *Aglaonema costatum* N.E.Br., *Syngonium angustatum* Schott, *Pistia stratiotes* L., *Xanthosoma helleborifolium* (Schott) Schott, and *Zomicarpella amazonica*) in this study. JSA junctions were completely included in the *ycf1* gene in four *Anubias* plastomes and in the other seven plastomes mentioned above in the JSB junctions. JLA junctions were highly conserved in four *Anubias* plastomes and other plastomes with the presence of *rpl2* and *trnH* genes, except in *Zantedeschia elliottiana* (with the presence of *rps19* and *psbA* genes) and *Anchomanes hookeri* (with the presence of *trnQ* and *psbK* genes) (Appendix A).

### 3.3. Repeats, Nucleotide Diversity, Codon Usage and RNA Editing Sites

The number of SSRs observed in four *Anubias* plastomes ranged between 365 (*A. heterophylla*) and 376 (*A. barteri* var. *nana*). Mononucleotide A/T repeats were the most abundant types of repeats in four *Anubias* plastomes (Figure 2A, Appendix A). Analysis of the long repeats (forward, reverse, palindromic, and complement repeats) showed high consistency in repeat number among four *Anubias* plastomes. Forward repeats were the most abundant types of repeats in four *Anubias* plastomes. Complement repeats were the most unusual types of repeats in *A. barteri* and *A. barteri* var. *nana*, while reverse repeats were the most unusual types of repeats in *A. heterophylla* and *A. hastifolia* (Figure 2B, Appendix A).

Interspecific variation among four plastomes of *Anubias* conducted in the online program mVISTA showed that the coding region was more conserved than the noncoding region (Figure 3). Furthermore, the nucleotide diversity (Pi) of four *Anubias* plastomes was analyzed. Sliding window analysis detects some regions with high Pi values, e.g., *trnL*-*CAA*-*ndhB*, *ycf1*-*ndhF*, and *rps15*-*ycf1* spacer regions with Pi values of 0.042, 0.025, and 0.017, respectively (Figure 4).

Codon usage of four *Anubias* plastomes were compared. The plastomes of *A. heterophylla*, *A. hastifolia*, *A. barteri*, and *A. barteri* var. *nana* exhibited 21,145; 21,358; 21,361; and 21,361 codons, respectively. Leucine (L) was the most frequently coded amino acid in all of the compared plastomes. Cysteine (C) was the least coded amino acid in the plastomes of *A. heterophylla* and *A. hastifolia*, while tryptophan (W) was the least coded amino acid in the plastomes of *A. barteri* and *A. barteri* var. *nana* (Figure 5). The highest RSCU values were 1.94, 1.93, 1.93, and 1.93, while the lowest values were all 0.29 in *A. heterophylla*, *A. hastifolia*, *A. barteri*, and *A. barteri* var. *nana* (Appendix A). Codon usage was biased toward adenine (A) and thymine (T) in all of the compared *Anubias* plastomes. RNA editing analysis showed similarities with respect to genes and the position of editing sites in the coding genes. We predicted RNA editing sites in 23 to 24 genes among four *Anubias* plastomes (Appendix A). The *matK* gene contained one RNA editing site in the plastome of *A. hastifolia* with the conversion of TCA to TTA. The *rpl2* genes contained ACG as a start codon instead of ATG in four *Anubias* plastomes.

### 3.4. Phylogenetic Analysis

In the phylogenetic analysis (Figure 6), the four representatives of *Anubias* formed a monophyletic group with strong support (BS = 100), in which *A. heterophylla* was sister to *A. hastifolia* (BS = 100), with *A. barteri* and *A. barteri* var. *nana* being sister to the clade of *A. heterophylla* and *A. hastifolia* (BS = 100). The genus *Anubias* was sister to *Montrichardia arborescens* (L.) Schott with robust support (BS = 99).

### 3.5. Selective Pressure Analysis

A total of 78 consensus protein-coding genes of four *Anubias* plastomes were selected to estimate the selective pressure. Four genes, *ccsA*, *matK*, *ndhF*, and *ycf4*, were identified to have undergone positive selection with ω values of 20.827, 21.224, 58.206, and 137.628, respectively (Table 2, Appendix A).

## 4. Discussion

### 4.1. Chloroplast Genome Features and Comparisons

Three complete plastomes of *Anubias* were obtained in this study. The comparative analysis of the plastomes within the genus was first reported. The plastomes of *A. hastifolia*, *A. barteri*, and *A. barteri* var. *nana* were extremely similar to the published *A. heterophylla* in terms of genome size, genome structure, and gene content, indicating that the plastomes of *Anubias* were relatively conserved. Similar genome structures and gene contents were also reported in other plastomes of Araceae [12,40,41]. In all analyzed plastomes, the regions with higher GC content are located in IR regions or coding sequences, and the same findings are found in previous studies [12,40,41].

The contraction and expansion of the IR borders are common evolutionary events in most angiosperms (e.g., monocots) [42]. In this study, a comparative analysis of IR borders revealed that the functional *ycf1* gene existed in the JSA junction and that the other pseudogene *ycf1* copy was located at the JSB junction among four *Anubias* plastomes. The same phenomenon was also found in seven analyzed plastomes within the family Araceae in this study and other angiosperm plastomes in previous studies [12,41].

SSRs, also known as microsatellites, have been widely applied as molecular markers for population genetics and species delimitation in aquatic plants [43]. SSR analysis in this study revealed that mononucleotide A/T repeats were the most abundant type of repeats in four *Anubias* plastomes, which was similar to the previous studies in other species [41,44]. SSRs of *Anubias* reported for the first time in this study can act as potential markers for genetic diversity studies of the genus. Our study also revealed the numbers of four types of oligonucleotide repeats, in which forward repeats were the most abundant types of repeats. These repeats may play an important role in the generation of substitutions and InDels, as previous studies of nuclear and cp genomes have revealed [45,46].

Codon usage bias patterns in genomes can be used to reveal phylogenetic relationships between organisms, horizontal gene transfers, and the molecular evolution of genes and identify selective forces [47]. RSCU values, as an important parameter, have been used for evaluating the codon usage bias degree [48]. A higher RSCU value (RSCU > 1) denotes the more frequently used codon in a gene, and a lower RSCU value (RSCU < 1) denotes the less frequently used codon [31]. High RSCU values of the codons are probably related to amino acid functions that avoid transcriptional errors in cp genomes [49]. Our study revealed that the highest abundance of the amino acid was leucine in four *Anubias* plastomes, which is similar to previous studies [41].

RNA editing is described as a posttranscriptional change in RNA nucleotides that alters a cytosine (C) to uridine (U) at a specific codon [50]. This is the key mechanism by which RNA maturation avoids incorrect mutations and enriches genetic information [51]. Our analysis revealed that the *ndhB* gene contained the most RNA editing sites, within 12 potential RNA editing sites, which is similar to previous reports in other species [44]. Additionally, we also observed that the *rpl2* gene contained ACG as a start codon instead of ATG in four *Anubias* plastomes, and this phenomenon was first reported in the maize plastome [52]. RNA editing mostly occurs in the first and second bases of codons, resulting in the conversion of hydrophilic amino acids to hydrophobic amino acids [53]. Research on RNA editing can improve our understanding of the gene expression and molecular evolution mechanisms of *Anubias*.

Saad et al. [4] revealed that the *matK* gene had shown high potential as DNA barcoding for selected *Anubias* species in their research. Our results also supported that the *matK* gene had a high variability in the coding region. However, the coding region was more conserved than the noncoding region, which is consistent with previous studies in other species [44]. Hence, three intergenic spacer regions (*trnL*-CAA-*ndhB*, *ycf1-ndhF*, and *rps15*-*ycf1*) with higher level of variation were selected in this study, which could be used for further development in applications such as DNA barcoding and phylogenetic reconstruction.

### 4.2. Phylogenetic Inference

Although the phylogenetic position of the genus among the monocots was previously known, the accuracy of the phylogenetic relationship needs to be confirmed with more evidence because of the insufficient sample size (21–23/75 genera in the subfamily Aroideae) [12,40]. In our study, we extended the taxon sampling to 27 genera within the subfamily Aroideae. Although the sampling size is still limited, our result was consistent with previous studies with robust support (BS = 99) [12,40]. The result was also supported by the same basic number of chromosomes (x = 24) shared by *Anubias* and *Montrichardia* [54]. In addition, the uncertain position of *Calla* L. and *Schismatoglottis* Zoll. & Moritzi in the phylogenies of Araceae reported in a previous study [12] was still unresolved in our study with low support values. We propose that future phylogenetic studies of Araceae should be focused on wider taxa sampling and nuclear genomes.

### 4.3. Adaptations to the Aquatic Environment

All of the *Anubias* species have adapted to the aquatic environment, indicating that this species might have evolved mechanisms to respond to different abiotic stresses that occur underwater, such as low light level, reduced carbon and oxygen availability, and mechanical damage through wave exposure. For example, *A. barteri* may adapt to low light level by increasing leaf area and chlorophyll content and to low O_2_ and CO_2_ concentration underwater through forming adventitious roots [55]. Considering the characteristics of their slow growth rate [7] and unusual morphological anatomy (e.g., no aerenchyma tissues differentiated in the adventitious root of *A. barteri*) [55], we speculate that *Anubias* possibly adopts the low-oxygen quiescence strategy (LOQS, an energy-saving mode) [56] to adapt to the aquatic environment. However, this needs to be confirmed by future studies.

With the exception of morphological, physiological and anatomical adaptations, some molecular adaptations (e.g., gene loss, gene positive selection) of aquatic plants could play an important role in the aquatic environment [57,58]. In this study, there was no gene loss other than four genes (*ccsA*, *matK*, *ndhF*, and *ycf4*) under positive selection with significant selective sites in *Anubias* plastomes. Previous studies have revealed the *ccsA* gene with positive selection in some aquatic plants such as *Oryza* L. [59], Zosteraceae [58], and some species of Lythraceae [60]. The *ccsA* gene is required for the biogenesis of c-type cytochromes at the step of heme attachment, and its function is linked to electron transfer in respiration and photosynthesis [61,62]. The *matK* gene has been reported to be under positive selection in some aquatic or hygrophilous plants (e.g., *Oryza*, *Lupinus* L.) [59,63]. This gene encodes an intron maturase that is involved in the splicing of group II RNA transcriptional introns, and its function is linked to plastid translation and photosynthesis [64,65,66]. Currently the *ndhF* gene under positive selection has not yet been reported in aquatic plants, but has been reported in some land plants, such as *Debregeasia* Gaudich. (Urticaceae) [67], *Rheum* L. (Polygonaceae) [68], and *Limonium* Mill. (Plumbaginaceae) [69]. The *ndhF* gene is a subunit of NADH-dehydrogenase, and its functions are linked to cyclic electron flow around photosystem I, essential for photosynthesis [70,71]. Some studies have reported the *ycf4* gene with positive selection in several plants, such as some species of Zingiberaceae [72], Lythraceae [60], and Orchidaceae [73]. The *ycf4* gene encodes a protein as a nonessential assembly factor for photosystem I, and it may have additional functions in chloroplasts deficient in photosystem II repair [74,75]. Further studies of *Anubias* plants chloroplast genomes should determine whether the positive selection of these genes is related to the adaptations to the aquatic environment.

## 5. Conclusions

In this study, three complete plastomes of *Anubias* were newly sequenced, and the structures of four plastomes within the genus were compared. Through comparative analyses, some important genetic features (such as IR contraction and expansion, SSRs, codon usage and RNA editing sites) were obtained. Three spacer regions (*trnL*-*CAA*-*ndhB*, *ycf1*-*ndhF*, and *rps15*-*ycf1*) with a high potential to be developed for DNA barcoding were found in the *Anubias* plastomes. Phylogenetic analysis showed that *Anubias* was sister to *Montrichardia* with robust support. Four genes (*ccsA*, *matK*, *ndhF*, and *ycf4*) were identified to have undergone positive selection. These results could provide valuable information for further studies on species identification and the evolutionary history of *Anubias,* especially its molecular adaptations to the aquatic environment. However, more species sampling of *Anubias* and other genera in the family Araceae is needed to facilitate our understanding of the phylogeny and evolutionary history within *Anubias* in the future.

## Figures and Tables

**Figure 1 genes-13-02043-f001:**
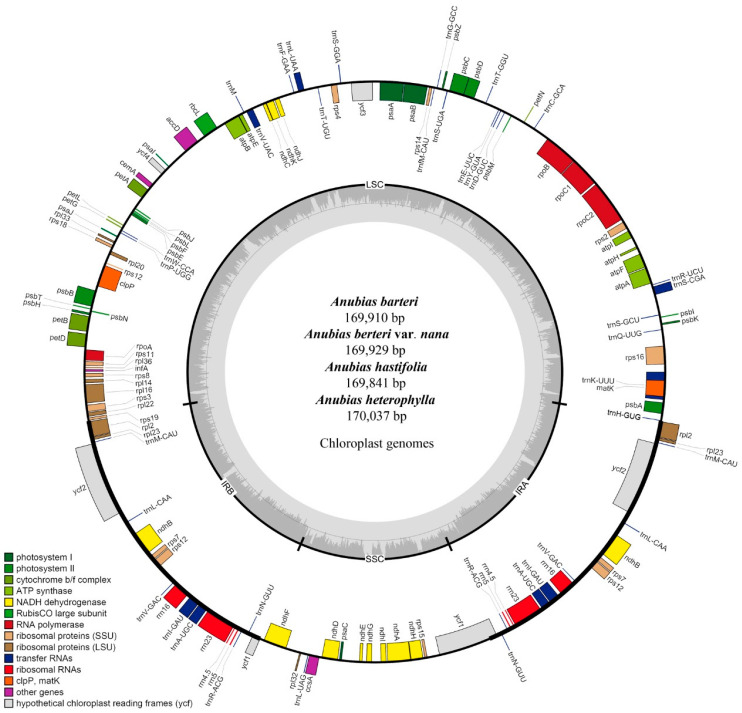
Circular map of chloroplast genomes of four *Anubias* plants. Genes shown inside of the inside layer circle are transcribed clockwise, whereas those genes outside of this circle are transcribed counterclockwise. The colored bars indicate the known protein-coding genes, tRNA, and rRNA. The darker gray area of the inner circle denotes the GC content, while the lighter gray area indicates the AT content of the genome. LSC, large single-copy; SSC, small single-copy; IR, inverted repeat.

**Figure 2 genes-13-02043-f002:**
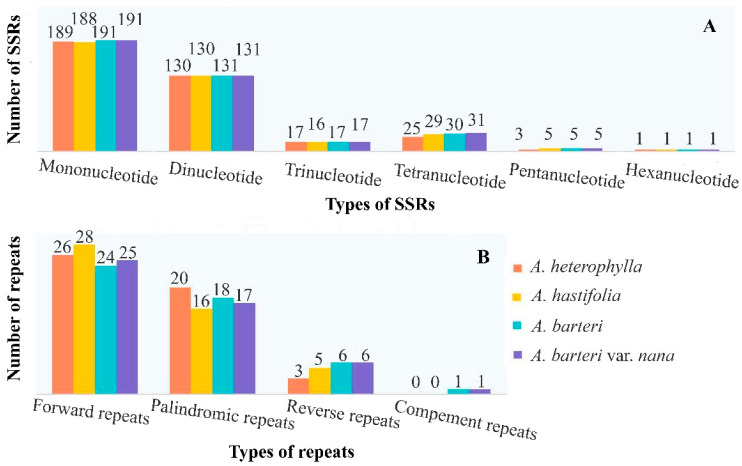
Four types of long repeats in four Anubias plastomes. (**A**) Number of SSRs; (**B**) Number of four types of long repeats.

**Figure 3 genes-13-02043-f003:**
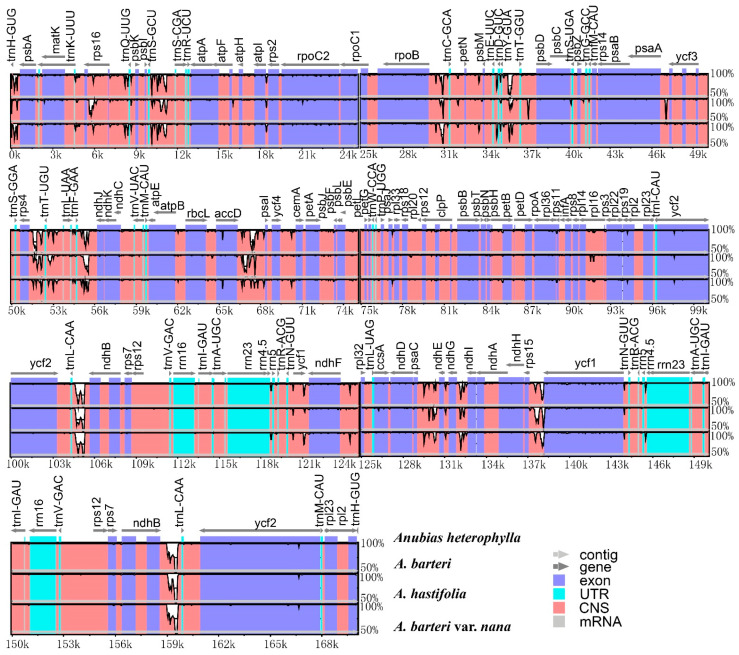
Alignment visualization of four *Anubias* plastomes with the mVISTA program.

**Figure 4 genes-13-02043-f004:**
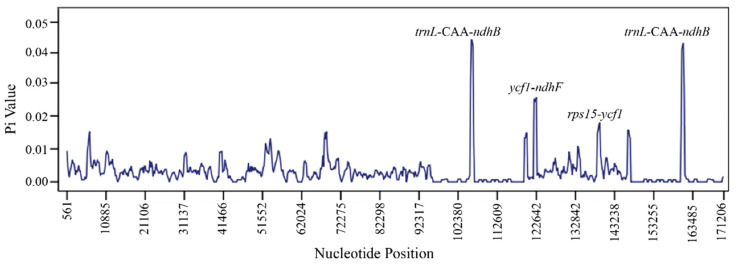
Nucleotide diversity and hotspot regions of four *Anubias* plastomes.

**Figure 5 genes-13-02043-f005:**
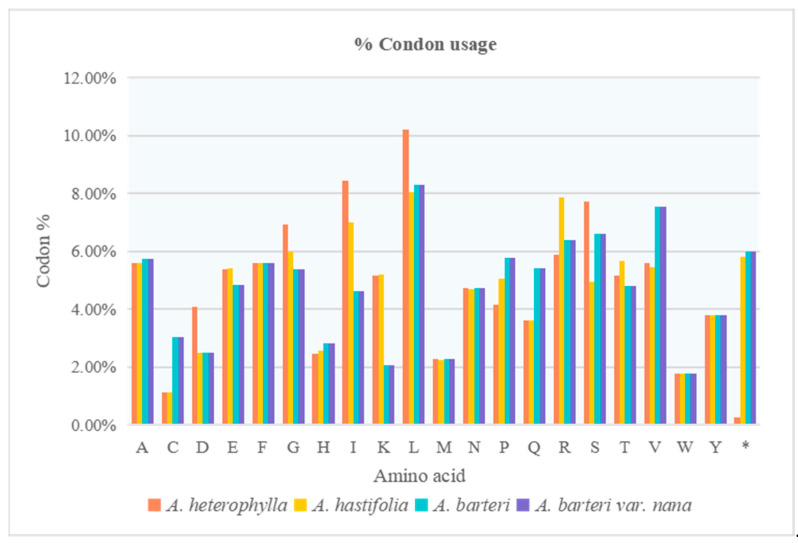
Codon Usage in four *Anubias* plastomes. * stop condon.

**Figure 6 genes-13-02043-f006:**
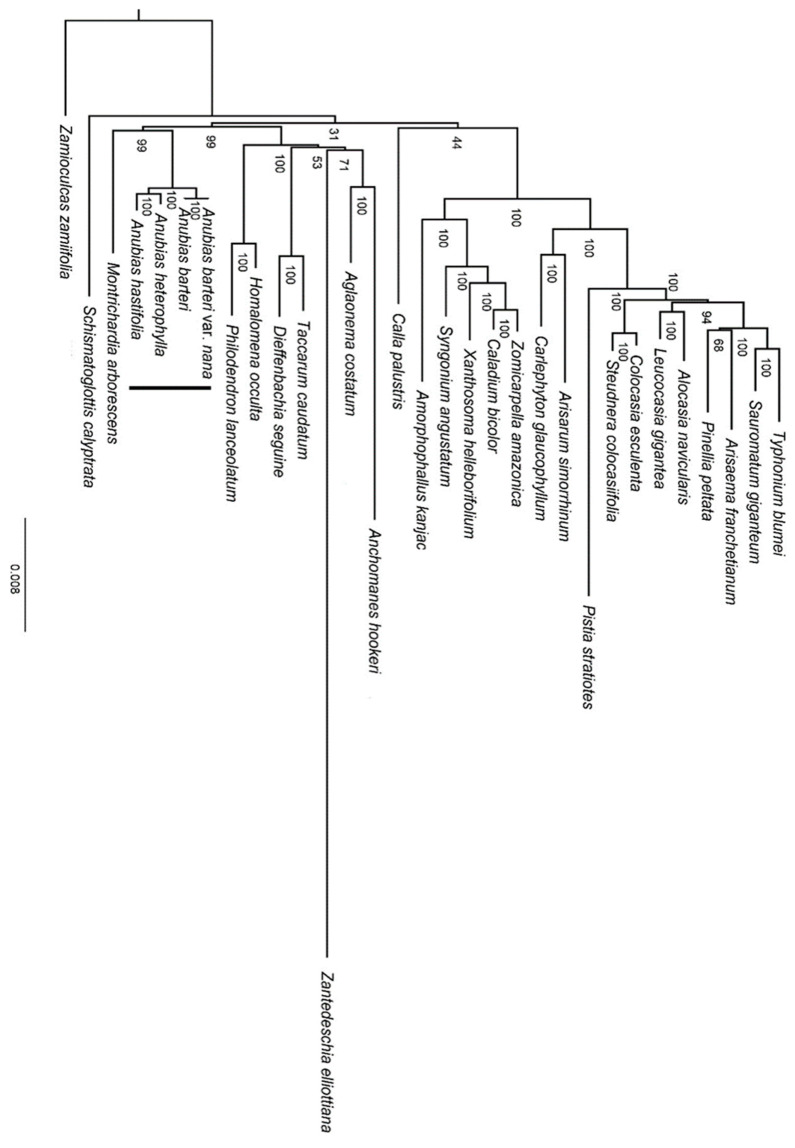
ML phylogenetic tree reconstruction of *Anubias* and other genera in Araceae based on 75 consensus CDSs.

**Table 1 genes-13-02043-t001:** List of genes identified in four *Anubias* plastomes.

Category of Genes	Group of Gene	Name of Gene
Self-replication	Ribosomal RNA genes	*rrn*4.5 ^×2^, *rrn*5 ^×2^, *rrn*16 ^×2^, *rrn*23 ^×2^
	Transfer RNA genes	*trnC*-GCA, *trnD*-GUC, *trnE*-UUC, *trnF*-GAA, *trnG*-GCC, *trnH*-GUG, *trnK*-UUU **, trnL*-UAA **, trnL*-UAG, *trnM*, *trnM*-CAU ^×2^, *trnP*-UGG, *trnQ*-UUG, *trnR*-UCU, *trnS-*CGA, *trnS*-GCU, *trnS*-GGA, *trnS*-UGA, *trnT*-UGU, *trnT*-GGU, *trnV*-UAC **, trnY*-GUA, *trnW*-CCA, *trnfM*-CAU, *trnA*-UGC *^,×2^, *trnI*-GAU *^,×2^, *trnL*-CAA ^×2^, *trnN*-GUU ^×2^, *trnR*-ACG ^×2^, *trnV*-GAC ^×2^
	Ribosomal protein (small subunit)	*rps2, rps3, rps4, rps7*×2, *rps8, rps11, rps12* **^,×2^, *rps14, rps15, rps16* *, *rps18, rps19*
	Ribosomal protein (large subunit)	*rpl2* *^,×2^, *rpl14, rpl16 *, rpl20, rpl22, rpl23* ^×2^, *rpl32, rpl33, rpl36*
	RNA polymerase	*rpoA, rpoB, rpoC1 *, rpoC2*
	Translational initiation factor	*infA*
Genes for photosynthesis	Subunits of photosystem I	*psaA, psaB, psaC, psaI, psaJ, ycf3* **, *ycf4*
	Subunits of photosystem II	*psbA, psbB, psbC, psbD, psbE, psbF, psbH, psbI, psbJ, psbK, psbL, psbM, psbN, psbT, psbZ*
	Subunits of cytochrome	*petA, petB *, petD *, petG, petL, petN*
	Subunits of ATP synthase	*atpA, atpB, atpE, atpF* *, *atpH, atpI*
	Large subunit of Rubisco	*rbcL*
	Subunits of NADH dehydrogenase	*ndhA* *, *ndhB* *^,×2^, *ndhC, ndhD, ndhE, ndhF, ndhG, ndhH, ndhI, ndhJ, ndhK*
Other genes	Maturase	*matK*
	Envelope membrane protein	*cemA*
	Subunit of acetyl-CoA	*accD*
	Synthesis gene	*ccsA*
	ATP-dependent protease	*clpP ***
	Component of TIC complex	*ycf1* ^×2^
Genes of unknown function	Conserved open reading frames	*ycf2* ^×2^

Note: ^×2^: Two gene copies in IR regions; *: With one intron; **: With two introns.

**Table 2 genes-13-02043-t002:** Positive selection sites in four *Anubias* plastomes.

Gene Name	Positive Sites
** *ccsA* **	90 Y 0.935, 91 F 0.843, 92 R 0.636, 198 H 0.937, 211 Y 0.598, 216 L 0.625
** *matK* **	37 L 0.630, 46 E 0.618, 94 F 0.607, 95 D 0.933, 214 R 0.660, 393 P 0.615, 459 P 0.936
** *ndhF* **	49 N 0.579, 53 V 0.566, 291 M 0.568, 463 Q 0.575, 571 D 0.568, 644 G 0.871
** *ycf4* **	129 G 0.886

## Data Availability

Data will be made available on request.

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
