# Peer review of "Comparative Analyses of Plastomes of Four Anubias (Araceae) Taxa, Tropical Aquatic Plants Endemic to Africa"

_genes, 2022, doi:10.3390/genes13112043_

Round 1

Reviewer 1 Report

In this manuscript, Li et al. sequenced and assembled the plastomes of A. barteri, A. barteri var. nana, and A. hastifolia. Then compared them with the published plastome sequences of A. heterophylla and other genera in the family Araceae. Furthermore, the phylogenetic relationships among Anubias and other genera in the Arum family were reconstructed based on consensus protein-coding genes. The main objectives were to 1) explore the size range and structure of the Anubias plastomes; 2) detect highly variable regions as the bases of developing molecular markers for species identification; 3) construct a phylogenetic tree for investigating the interspecific relationships within Anubias, as well as the relationships among Anubias and other genera in Araceae; 4) identify the protein-coding genes under positive selection within the four plastomes of Anubias. I did not actually recognize any real limitations of the manuscript. 

Author Response

Dear Reviewer,

Thank you very much for your time involved in reviewing the manuscript and your very encouraging comments on our study.

Sincerely,

Authors: Li Li, Changkun Liu, Kunpeng Hou, Wenzhe Liu *

Reviewer 2 Report

The main aim of this manuscript is to conduct comparative analysis of plastome of four species of Anubias. The genus is rather poorly represented in the GenBank, and obtaining information about the organization of chloroplast genomes seems to be quite important. 

Methodologically, the work was done quite well, but the problem statement and the interpretation of the results raise many questions. 

It is necessary to explain the choice of species included in the analysis. Whether these species are morphologically well-separated, or vice versa, attempts to distinguish between them cause the greatest difficulties. 

Please complete the vouchers with information about collection date and collector. 

How confident are the authors in the species of the studied specimens, since there are frequent cases of erroneous identification of plants grown in culture, including in botanical gardens? 

One of the tasks of the work is the search for molecular markers in the plastid genomes to better species identification. Four regions were found that differ from those previously identified. This contradiction needs to be clarified. 

Also authors aimed to understand the evolutionary history of Anubias. The phylogenetic position of the genus among the monocots was previously known, and further clarification is required to explain whether the expansion of the sampling could affect the relationships of the genus with other members of Araceae. 

Another problem posed by the authors is to understand the mechanism of adaptation of Anubias to the aquatic environment. This problem is mentioned in the Abstract and Discussion, but not in the Introduction. It is also necessary to describe in more detail the life form of the studied species, specifying whether they are obligate aquatic plants or not. An important factor, not mentioned in the article, which distinguishes the aquatic environment from the terrestrial one, is the absence of stress caused by a sharp change in light intensity. In this regard, the analysis of the functionality of the ndh genes is necessary. Many references in section 4.3. (Adaptations to aquatic environment and the positive selection genes) refer either to the function of genes or to examples of plants from completely different habitats. Thus, the Introduction and section 4.3. are confusing and should be rewritten. 

The authors carry out a comparative analysis of plastomes of Anubias with other members of the family. However, there is no comparison with its closest relative, Montrichardia, to elucidate whether any structural rearrangements exist in addition to nucleotide substitutions. 

Please insert the authors at the first mention of species names.

Author Response

Dear Reviewer,

Thank you very much for your time involved in reviewing the manuscript and your very valuable comments and suggestions on our work.

Round 2

Reviewer 2 Report

The section 4.3. (Adaptations to the aquatic environment and the positive selection genes) seems to be in need of some work, especially the relationship between positive selection of the genes and the aquatic habitat conditions. This part of the article has a very superficial discussion, as it does not make comparisons with objects with similar ecologies. Many of the examples, in which cases of positive selection of individual genes have also been noted, refer to plants that have a completely different ecology, even if they grow in humid places. Thus, the examples given do not convince us that the observed positive selection is associated precisely with adaptation to an aquatic lifestyle. Rather, one might think that not only in the organization of the plants themselves, their anatomy or morphology, but also in the organization of the chloroplast genome, no changes associated with aquatic habit have occurred. If so, then this could also be a very important conclusion.

It is also necessary to note that not only positive selection, but also gene loss can act as adaptations to an aquatic environment (see, for example, Peredo et al. 2013, Plos One 8(7), e68591).

I encourage the authors to either cite the relevant literature and rewrite this section in more meaningful and detailed ways, or ignore the topic.

Author Response

Dear Reviewer,

Thank you very much for your efforts in reviewing our revised manuscript and your valuable comments and constructive suggestions.

In the remainder of this letter, we discuss each of your comments individually along with our corresponding responses.
